# Instruments used to measure knowledge and attitudes of healthcare professionals towards antibiotic use for the treatment of urinary tract infections: A systematic review

**Angela Kabulo Mwape**[ID]*, **Kelly Ann Schmidtke**[ID], **Celia Brown**[ID]

Division of Health Sciences, Warwick Medical School (WMS), University of Warwick, Coventry, United Kingdom

* angela.mwape@warwick.ac.uk

## Abstract

### Background

Urinary tract infections (UTIs) are the second most common condition (after upper respiratory tract infections) for which adults receive antibiotics, and this prevalence may contribute to antibiotic resistance. Knowledge and attitudes have been identified as potential determinants of antibiotic prescribing behaviour among healthcare professionals in the treatment and management of UTIs. An instrument that captures prescribers' baseline knowledge of and attitudes towards antibiotic prescribing for UTIs could inform interventions to enhance prescribing. The current systematic review evaluates the psychometric properties of instruments already available and describes the theoretical constructs they measure.

### Methods

Five electronic databases were searched for published studies and instruments. The Consensus-based Standards for the selection of health status Measurement Instruments checklist was used to assess the psychometric quality reporting of the instruments. The items included in each instrument were mapped onto the theoretical constructs underlying knowledge and attitudes using a mixed-theoretical model developed for this study.

### Results

Fourteen studies met the review inclusion criteria. All instruments were available for review. None of the instruments had all the psychometric properties evaluated. Most of the instruments sought to identify knowledge and/or attitude factors influencing antibiotic prescribing for UTIs rather than to measure/assess knowledge and attitudes.

### Conclusions

Few instruments for the assessment of knowledge and attitudes of healthcare professionals towards antibiotic use and UTI treatment are available. None of the instruments underwent the full development process to ensure that all psychometric properties were met.

**Data Availability Statement:** All relevant data are within the manuscript and its Supporting Information files.

**Funding:** The current project was supported by the National Institute for Health Research (NIHR) Applied Research Centre (ARC) West Midlands, grant number NIHR200165. The views expressed are those of the author(s) and not necessarily those of the NIHR, ARC, or the Department of Health and Social Care. The funders had no role in the design of the study and collection, analysis, and interpretation of data and in writing the manuscript.

**Competing interests:** The authors have declared that no competing interests exist.

Furthermore, none of the instruments assessed all domains of knowledge and attitudes. Therefore, the ability of the instruments to provide a robust measurement of knowledge and attitudes is doubtful. There is a need for an instrument that fully and accurately measures the constructs of knowledge and attitude of healthcare professionals in the treatment of UTIs.

## Introduction

Heterogenous definitions of Urinary tract infections (UTIs) exist among researchers which are not always clear [1]. In broad terminology, UTIs can be defined as an infection involving any part of the urinary tract, namely the urethra (urethritis), bladder (cystitis), ureters, or kidneys. UTIs are mainly caused by microorganisms such as Escherichia coli (E. coli), Staphylococcus saprophyticus, and Enterococcus faecalis [2]. E. coli remains the predominant uropathogens accounting for 70–95% of cases [3]. UTIs can be classified into five different categories as shown in Table 1. Research by Foxman [4] and Flores-Mireles et al. [5] provides additional information on the risk factors of UTIs as well as their pathogenesis which is further explained in S1 Fig.

Urinary tract infections are the second most common condition (after upper respiratory tract infections) for which adults receive antibiotics [6], and account for 1 to 3% of primary care consultations in the United Kingdom [7]. While antibiotic prescriptions for UTIs will generally improve patients' health, not all healthcare professionals adhere to treatment guidelines [8]. The consequences of deviations from guidelines include unnecessary side effects for the patient and increased antibiotic resistance in the community [9].

Healthcare professionals have been offered various behavioural interventions aimed at addressing the inappropriate prescribing of antibiotics [10–19]. Systematic reviews by Wilkinson et al. [19] and Davey et al. [11], highlight key behavioural knowledge and antimicrobial stewardship interventions that can be used to address inappropriate prescribing behaviour among healthcare professionals. Knowledge interventions include audits, education, and stewardships. Audits involve reviews of prescribing patterns according to guidelines. Education

**Table 1. Classification of UTIs.**

| | |
|---|---|
| Uncomplicated UTIs | Acute, sporadic, or recurrent lower (uncomplicated cystitis) and /or upper (uncomplicated pyelonephritis) UTI, limited to non-pregnant women with no known relevant anatomical and functional abnormalities within the urinary tract or comorbidities. |
| Complicated UTIs | All UTIs which are not defined as uncomplicated. Meaning in a narrower sense UTIs in a patient with an increased chance of a complicated course: i.e., all men, pregnant women, patients with relevant anatomical or functional abnormalities of the urinary tract, indwelling urinary catheters, renal diseases, and/ or with other concomitant immunocompromising diseases such as diabetes. |
| Recurrent UTIs | Recurrence of uncomplicated and/or complicated UTIs, with a frequency of at least three UTIs /year or two UTIs in the last six months. |
| Catheter-associated UTIs | Catheter-associated urinary tract infection (CA-UTI) refers to UTIs occurring in a person whose urinary tract is currently catheterised or has had a catheter in place within 48 hours. |
| Urosepsis | Urosepsis is defined as life-threatening organ dysfunction caused by a dysregulated host response to infection originating from the urinary tract and/or male genital organs. |

From: The European Association of Urology (EAU) (2021). Guidelines on Urological Infections. Bonkart, G. B., R. Bruyere, F. Cai, T. Geerlings, S.E. Koves, B. Schubert, S. Wagenlehner, F (ed.). Netherlands.

involves training sessions, courses, and role plays for which feedback is provided via meetings, reports, group discussions, handbooks, and ward posters [20, 21]. Antimicrobial stewardship interventions include the combination of policy review and change (e.g., development of guidelines, incentives/disincentives, and new targets) as well as the creation of antimicrobial/ antibiotic resistance committees. Other interventions may also include structural changes (introduction of a new diagnostic test or clinical algorithm to guide prescriptions) or a bundle of the aforementioned types (mix of different interventions). Despite the abundance of interventions, the effectiveness of these interventions remains questionable as deviation from evidence-based guidelines among healthcare professionals continues [8]. Although not specific to UTIs, a systematic review by Nair et al. [18] and Tonkin-Crine et al. [14] suggests that effective interventions are multifaceted in their approach and addressed all concerns of healthcare professionals. The World Health Organisation (WHO) agrees and has acknowledged that effective interventions must also fill existing gaps in healthcare professionals' knowledge and attitudes [22].

Several instruments exist to assess healthcare professionals' knowledge and attitudes towards antibiotic prescribing in general, but they lack evidence of meeting psychometric properties [23]. Five highly regarded psychometric properties are that the instrument provides a (1) valid and (2) reliable measurement, that is (3) acceptable, (4) feasible, and has the (5) potential for future educational impact [24, 25]. More recent work, for example, that of Teixeira Rodrigues et al. [17] and López-Vázquez et al. [26], aims to remedy this deficiency towards antibiotic prescribing in general rather than specifically for UTIs. An instrument looking at UTI prescribing specifically may be important, as this is one of the most common conditions for which antibiotics are prescribed. In addition to specifying the condition, Alumran et al. [23] note the importance of developing fully validated instruments within specified target populations. Therefore, the current review seeks to draw out information about instruments designed to measure prescribers' knowledge and attitudes towards the prescribing of antibiotics for UTIs.

Although the constructs of Knowledge, Attitudes and Practice (KAP) are the key factors relating to measurement studies that utilise the WHO's recommendations for developing KAP surveys [27], the construct of practice was excluded from this review. This is because practice can be measured using routine prescribing data or physician observation in the treatment of patients whereas knowledge and attitude are constructs that are not directly observable and are difficult to measure. To measure knowledge and attitude, as well as to understand their influence on practice, it is important to ensure that any measurement instrument is based on well-defined content areas of knowledge and attitude and has sound psychometric properties.

Previous explanatory models for health behaviour, including antibiotic prescribing, suggests that knowledge and attitudes are multifaceted constructs [28]. We adapted a mixed theoretical model composed of Michie et al. [29]'s knowledge construct, and that of Teixeira Rodrigues et al. [30]'s attitudes construct. In our adapted model, we conceptualise physicians' knowledge in terms of the knowledge of condition, knowledge of scientific rationale, knowledge of procedure, and knowledge of the task environment [29]. We conceptualise attitudes in terms of physicians' complacency, fear, ignorance, indifference, and responsibility of others [30] (Tables 2 and 3).

The relationship between knowledge, attitudes, and practice can also be modified by exogenous factors affecting prescribing behaviour such as patient characteristics [18]. These characteristics include patients' socioeconomic status, age, and ethnicity [19]. Therefore, healthcare professionals' knowledge and attitudes towards these characteristics will also exert some influence on prescribing behaviours as shown in our proposed mixed-model (S2 Fig).

The primary aim of the present systematic review is to evaluate the psychometric properties of available instruments that assess healthcare professionals' knowledge and attitudes towards antibiotic prescribing for the treatment of UTIs. The secondary aim is to describe the knowledge and attitudes constructs assessed by those instruments.

**Table 2. Conceptualisation of knowledge.**

| Key term | Domain | Definition |
|---|---|---|
| **Knowledge (the awareness of the existence of UTIs)** | Condition | Knowledge of the condition of UTIs in terms of the diagnosis, investigations, clinical presentation, treatment/management, knowledge of local AMR levels and antibiotic resistance |
| | Scientific Rationale | Knowledge of scientific rationale of UTIs e.g., based on evidence-based guidelines, national AMR health policies |
| | Procedure | Knowledge of the relevant procedures undertaken in the diagnosis and treatment of UTIs e.g., relevant diagnostic tests and antibiotic treatment as per evidence-based guidelines |
| | Task of environment | Knowledge of the external influence on clinician's decision making when diagnosing and treating UTIs e.g. clinical decision-making aids, local AMR patterns, the influence of other staff members |

Adapted from: Michie, S., Johnston, M., Abraham, C., Lawton, R., Parker, D., Walker, A. & "Psychological theory" group. 2005. Making psychological theory useful for implementing evidence-based practice: a consensus approach. Quality & safety in health care, 14, 26–33.

## Methods

This review follows the Preferred Reporting Items for Systematic Reviews and Meta-Analyses (PRISMA) (S3 Fig) [31]. This review was registered with PROSPERO (Registration number CRD42021246369 available on the link: https://www.crd.york.ac.uk/prospero/display_record. php?RecordID=246369.).

### Information sources

Five electronic databases (MEDLINE via Ovid, PubMed, EMBASE, Web of Science and PsycINFO) were searched on the 5[th] of March 2021.

**Table 3. Conceptualisation of attitudes.**

| Key term | Domain | Definition |
|---|---|---|
| **Attitude** | Complacency | Attitude that motivates the prescribing of antibiotics to fulfil professionals' perceptions of their patients'/parents' expectations. |
| | Fear | Attitude relating to fear of possible future complications in the patient. Fear of losing patients (to other providers). |
| | Ignorance | Lack of relationship between overprescribing and antibiotic resistance, linked to a lack of knowledge. |
| | Indifference | Lack of motivation to feel positively or negatively inclined to the problem of antibiotic prescribing. |
| | Responsibility | Attitude underlying the belief that responsibility for generating antibiotic resistance lies with other professionals. |
| | Confidence | Term that seeks to describe the self-reliance felt by physicians when prescribing antibiotics. This attitude may be defined as the level of confidence felt by physicians when deciding whether to prescribe any given therapy including antibiotics, based on the maxim 'never change a winning practice' and the negative attitude towards single-dose antibiotic regimen as a result of a lack of confidence in their effectiveness |

Adapted from: Texeira Rodrigues, A., Roque, F., Falcao, A., Figueiras, A. & Herdeiro, M. T. 2013. Understanding physician antibiotic prescribing behaviour: A systematic review of qualitative studies. Int J Antimicrob Agents, 41, 203–12.

## Eligibility criteria

To perform a highly inclusive search, the key search terms were refined in consultation with a senior librarian from the University of Warwick. To capture the relevant studies, we applied a search strategy that ensured the correct use of Boolean operators, truncation, and medical subject headings (MeSH). The search strategy (Table 4) was based on keywords in five domains which were all combined using the "AND" operator: (i) overall topic of interest–antibiotic resistance, (ii) construct of interest–knowledge and attitudes, (iii) type of measurement–instruments, (iv) target population–healthcare professionals, nurse, doctor, and (v) therapeutic area of interest—UTIs.

No restrictions were imposed on the design of the study, nor the psychometric properties reported (as we wanted to include instruments where no such data were reported). The full inclusion and exclusion criteria are shown in S1 Table. The full search strategy for MEDLINE database is shown in the (see S4 Fig).

In addition to the defined inclusion and exclusion criteria, the selection criteria was adapted from Carlson et al. [32], where the systematic review was restricted to empirical studies published in peer-reviewed journals reporting original research. We searched the bibliographies of retrieved articles and published reviews for additional studies. Studies had to be available in full text and published in the English language. For the purpose of this review, "instruments" are defined as surveys, questionnaires, tools, or scales which contain individual items that are answered or scored using predefined response options. "Constructs" are defined as the broad attributes or characteristics which these items (usually grouped into domains) are attempting to assess. The constructs of knowledge and attitudes were chosen to align with our proposed model (see S2 Fig), to comprehensively capture those individual factors identified in previous systematic reviews as influencing prescribers' medication choices [29, 30]. S2 Table shows the key terms included in the study eligibility criteria.

## Study selection

The search results from each database were saved in RIS text format and uploaded onto End-Note reference manager version X19 [33] and then uploaded onto Rayyan software where duplicates were removed [34]. Two reviewers independently screened the titles and abstracts of the potentially eligible studies (AM and CB) using the Rayyan software. Disagreements were resolved by consensus discussions considering the full texts, in the full-text review stage. The full-text review was carried out by one reviewer (AM), and independently cross-checked by another reviewer (CB).

**Table 4. Search strategy.**

| | |
|---|---|
| #overall topic of interest | (Drug resistance, microbial OR drug resistance, multiple, bacterial OR antimicrobial resistance OR drug resistance, bacterial) OR ((antimicrobial" OR antibiotic* OR antibacterial*) AND (resistan* OR misuse OR overuse OR mis-use OR over-use OR inappropriate)) |
| #construct of interest | (Health Behaviour OR health knowledge, attitudes, practice, OR attitude to health) OR (attitude* OR perception* OR belief* OR opinion* OR thought* OR feeling* OR view or views or experience*) |
| #type of measurement | (Instrument* or tool OR tools OR survey* OR measure*OR psychometric) |
| #target population | (Healthcare professionals OR healthcare workers OR healthcare providers or physician or nurse or doctor or general practitioner* OR gp OR family doctor or primary care or primary healthcare or family practice or doctor* or healthcare professional or physician*) |
| #therapeutic area of interest | (Urinary tract infection* OR uti* OR tract infection* OR urinary infection* or bladder infection*) |

## Data extraction

A data extraction form following Cochrane's guideline for the conduct of systematic reviews was developed and used to identify the relevant characteristics and information of included studies [35]. Due to the focus of the review, only aspects of the studies relating to the measurement instruments were extracted. To increase the accuracy of our data extraction, the information extracted was reviewed by all members of our interdisciplinary team, including an expert in psychometrics (CB) and psychology (KAS). The following general information was extracted from each study: author(s) names, publication year, number and type of respondents, sampling strategy, strategy design, therapeutic area (e.g., uncomplicated UTI, Pyelonephritis etc.), how questions were asked, method of data collection, outcome assessment, and findings. For each instrument described, data were extracted, to describe the specific construct of knowledge and attitudes considered according to the model described in the introduction.

Where one or more psychometric properties was assessed, the Consensus-based Standards for the selection of health Measurement Instruments (COSMIN) checklist [36] was used to evaluate the psychometric property(ies) of the included instrument. The COSMIN checklist is a tool designed for the assessment of both the risk of bias [36, 37] and the quality criteria for health instruments [38]. The risk of bias was not assessed, because the systematic review was non-interventional. The quality criteria for measurement properties were assessed. Due to the lack of specific tools for appraisal of knowledge and attitude instruments, the COSMIN tool was considered appropriate for this review as it covers an even wider range of aspects than only those relevant for educational interventions [36]. It has been widely used in the quality assessment of instruments [39]. The COSMIN tool provides separate checklists (referred to as boxes) for each type of measurement property, for example, box A is for internal consistency, box B for reliability and so forth. Boxes A-H are for different types of measurement properties and have their associated quality questions on whether a study on a specific measurement property meets the standards for good instrument quality. See Mokkink et al. [40] for a full explanation of the COSMIN checklist. From the measurement properties in the COSMIN tool, those covering validity and reliability were used in this study. Criterion validity was not rated as no gold standard instrument exists to assess knowledge and attitudes [23]. However, the COSMIN checklist only enables a critique of the validity and reliability properties of psychometric properties. Therefore, the psychometric properties of acceptability, feasibility, and potential for educational impact [25] were added as properties for evaluation using questions adapted from Beattie et al. [41]'s scoring criteria (S3 Table).

Two steps were undertaken in the evaluation of each psychometric property. AM rated each psychometric property and KAS checked the ratings. Firstly, the methodological quality of how each measurement property was being assessed within each study was evaluated. To evaluate the methodological quality of each measurement property, a 4-point scoring system was used: 'inadequate' when there was evidence that the methodological quality was not adequate, 'doubtful' if the methodological quality was in doubt, 'adequate' when the relevant information was not fully reported but adequate quality can be assumed, and 'very good' when there was evidence of adequate methodological quality as described by the COSMIN tool [36]. Where answers to checklist questions were of variable rating within one box (i.e., some very good, some inadequate), the overall score was determined by taking the lowest rating of any item. In other words, the "worst score count" [37]. For example, if one item in the box for the domain "reliability" is scored inadequate, the methodological quality of the assessment of reliability in that study is rated as inadequate. An inadequate score on any item is thus considered to represent a fatal flaw [37].

The psychometric properties of acceptability, feasibility and potential for educational impact were rated using Beattie et al. [41]'s scoring matrix. This was achieved using the star ratings: excellent (****), good (***), fair (**) and poor *.

The second step undertaken in the evaluation of each psychometric property quality involved the quality of the results of the psychometric property reported. This was carried out using the Quality criteria for Measurement Properties devised by Terwee et al. [42] (see S4 Table). Results were rated as positive (+), indeterminate (?), negative (−) or mixed (+-) according to the quality criteria for each measurement property. For example, positive ratings for internal consistency are given, using Terwee et al. [42] criteria, if Cronbach's alpha is $\geq$0.70. Studies with Cronbach's alpha results of <0.70 would be categorised as negative; or where Cronbach's alpha was not determined, the result would be categorised as indeterminate. A full explanation, with justification for all COSMIN criteria results, is available from Terwee et al. [42].

### Data synthesis

A narrative approach was used to synthesise the findings. As noted above, the specific constructs of knowledge and attitudes (Tables 2 and 3) assessed by each instrument were extracted by reviewing the content of each instrument (i.e., the actual questions being asked). The instrument items were organised using a grid. Gaps were identified in the grid to highlight the influences not reported in the literature.

## Results

### Study selection

Based on the titles and abstracts of the 2429 articles eligible for consideration, 29 articles were eligible for full-text screening (see PRISMA flow diagram in Fig 1). The level of agreement between the two reviewers for all 2429 articles was compared to check for inter-rater reliability, which gave a Kappa score of 0.66, 95% CI [0.53 to 0.79], indicating substantial agreement between the two reviewers [43]. The results for the consensus between the two reviewers are shown in S5 Table.

From the 29 articles at full-text screening, four further articles were identified through other systematic reviews: After reviewing the full texts of all 33 articles, 15 articles were excluded. Fourteen studies met the review's inclusion criteria as shown in the PRISMA flow diagram below in (Fig 1). Each study used a different instrument to assess knowledge and/or attitudes.

### Study characteristics

The characteristics of the included studies are summarised in Tables 5 and 6. All studies and the instruments they contained were freely available to the research team. Study characteristics such as cohort descriptors and sample sizes are in Table 5. Respondents were predominantly physicians and nurses across all studies. Those studies reporting on the psychometric properties of the instruments they employed (validity and reliability, acceptability, feasibility, and potential for educational impact) are summarised in Table 6.

### Psychometric properties

Tables 7 and 8 shows the summary table of quality of the psychometric properties evaluated within each study adapted COSMIN, Terwee scale scores and Beattie et al. [41]. The results for each property are also presented.

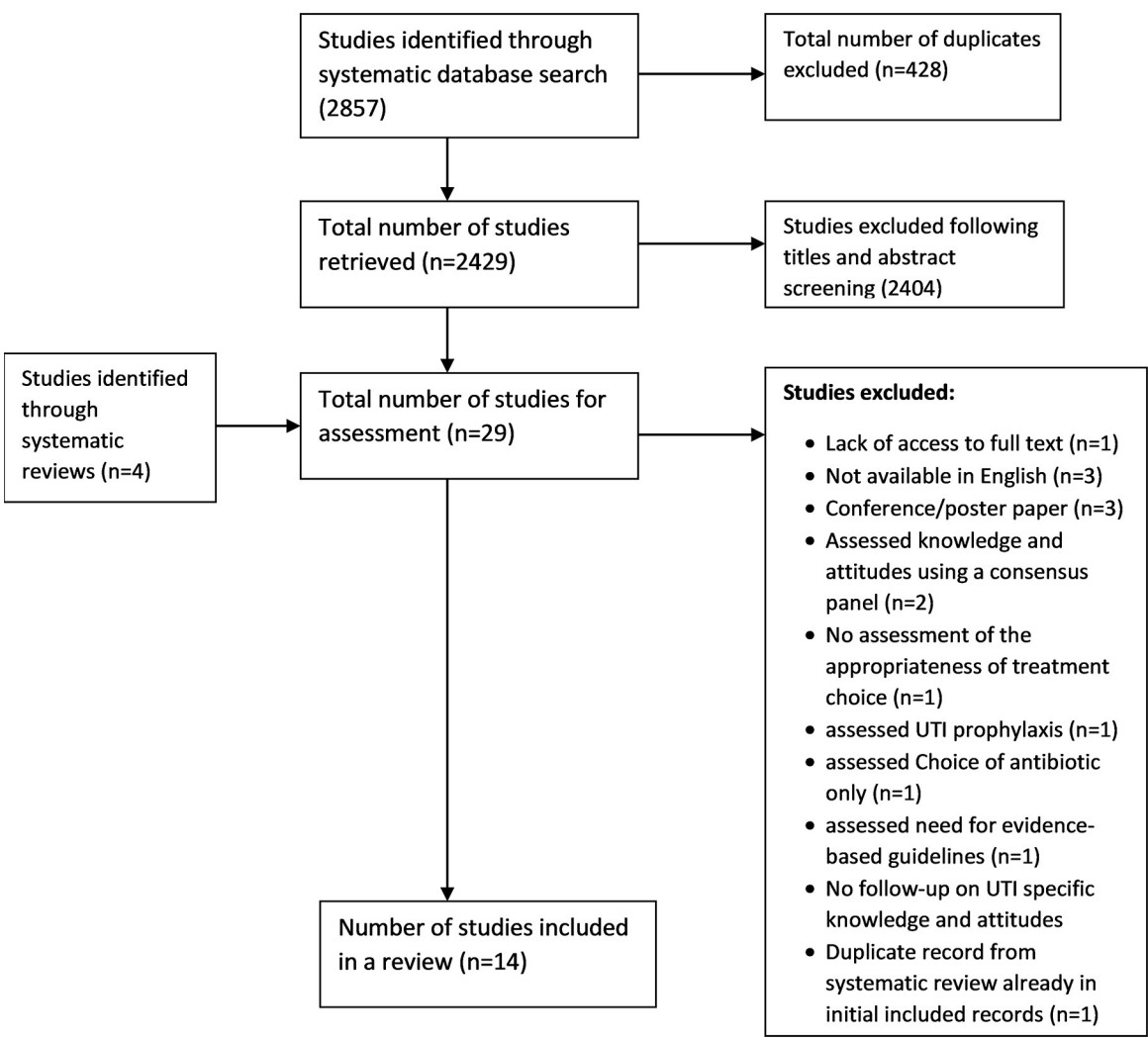

**Fig 1. PRISMA flow diagram.**

**Validity.** To demonstrate adequate validity, the instrument should undergo early (content and face validity), middle-stage (construct validity) and final stage validation (criterion validity). None of the studies attempted a full validation process of their instruments. Four studies attempted to validate their scales by pilot testing the scale [46, 47, 51, 56], which contributed to the assessment of the face validity of the instrument. Two other studies assessed content validity using expert opinions [49, 51]. Only half of the studies mentioned the development and validation process associated with their instrument: one instrument was adopted from other studies [46]; one was modified from the Minnesota Department of Health's Antibiotics Stewardship Program Toolkit for Long-Term Care Facilities [50]; two were developed by the team of authors who had expertise in infectious disease management [48, 49]; another instrument's study team included urologists [51]; another instrument's study team included lecturers in health sciences and psychiatry [55]; and the last instrument's study team included pharmacists and medical doctors [56].

The instrument with the most promising evidence for face and content validity was by Hale et al. [50] as it had more evidence of the type of professionals included in the validation

**Table 5. Study characteristics.**

| Author(s), Year | Study title | Number of respondents and country | Sampling strategy (% response rate, where reported) | Study Design and method | Therapeutic area | Method of data collection | Question format | Number of questions | Number of items |
|---|---|---|---|---|---|---|---|---|---|
| [44] | Empirical treatment of uncomplicated urinary tract infection by community pharmacist in the Eastern province of Saudi Arabia | 88 Pharmacist, Saudi Arabia | convenience sampling (not reported) | Cross-sectional mixed methods | Acute uncomplicated Urinary Tract Infections | simulated client method (SCM) with open ended questions | open ended simulated patient scenario with acute uncomplicated UTI to assess attitude and a quantitative survey to assess behaviour in terms of number of prescriptions given | 1 patient case scenario of which there was 1 question about attitudes with an open-ended response option. | 1 patient case scenario of which there was 1 question about attitudes with open-ended response |
| [45] | Failure to implement hospital antimicrobial prescribing guidelines: a comparison of two UK academic centres | 316 physicians, U.K | not reported | Cross-sectional mixed methods | general UTIs | vignettes followed by open-ended questions | UTI scenario with behavioural questions followed by a question about the major influences on those decisions, a question about resources access to aid prescribing decisions and lastly, a question about the physicians clinical grade | 2 patient case scenarios on community acquired pneumonia and UTI respectively, 1 question that would reveal knowledge information "what initial antimicrobial route, dose and frequency" 2 questions that would reveal attitude information: "What are the major influences on your decisions in above questions?" "Should this patient be admitted into hospital" and influence of sociodemographic factors such as clinical grade | 4 open ended questions following the case scenario |
| [46] | The impact of law enforcement on dispensing antibiotics without prescription: a multi-methods study from Saudi Arabia | 116 community pharmacists, Saudi Arabia | convenience sampling (116/116 = 100%) | cross-sectional self-administered quantitative survey and a simulated client method (SCM) | Pharyngitis and UTIs | simulated client method (SCM) with closed answered questions (Yes/No) | knowledge was yes no don't know before and after the laws implementation which was in regard to knowledge and perceptions of CPs towards DAwP with no specific DAwP questions for UTIs, then simulated patient scenario assessed whether an AB was dispensed without prescription, CP communicated with patient on adherence to course and reason behind refusal of DAwP: health related and regulations | 10 antibiotic/resistance knowledge questions and one simulated patient scenario about measuring attitudes towards dispensing without a prescription | 10 items on knowledge and perceptions towards DAwP; simulated patient requesting for medication on behalf of sister with UTI symptoms with 4 item audit criteria on dispensed antibiotic without prescription, educated on importance of adherence and completion of antibiotic course, how many times it took to dispense antibiotics and reason behind refusing antibiotics dispensing. |

*(Continued)*

**Table 5.** (Continued)

| Author (s), Year | Study title | Number of respondents and country | Sampling strategy (% response rate, where reported) | Study Design and method | Therapeutic area | Method of data collection | Question format | Number of questions | Number of items |
|---|---|---|---|---|---|---|---|---|---|
| [47] | Awareness of antibiotic resistance and antibiotic prescribing in UTI treatment: a qualitative study among primary care physicians in Sweden | 20 GPs, Sweden | Purposive sampling (not clear) | cross-sectional qualitative face to face semi-structured interviews | uncomplicated UTIs | face to face semi-structured interviews with open-ended probing questions performed by one of the authors at GPs workplace or home | Interview based to assess treatment, effect of treatment and resistance | 4 questions in 4 areas (introduction, treatment, effect of treatment and resistance) followed by an average of 5 questions within each area | 5 open-ended questions with 5 to 7 probing questions on treatment of patients with acute uncomplicated UTIs |
| [48] | A survey of the antibiotic prescribing practices of doctors in an Australian Emergency Department | 60 Doctors, Australia | not reported (58/89 = 65%) | cross-sectional emailed quantitative survey based on two hypothetical case scenarios | uncomplicated UTI and severe pyelonephritis | vignettes collected online using survey monkey | patient case scenario | 10 questions. Participants responded to 5 hypothetical cases, 2 of which were related to antibiotics for UTIs. After responding to all 5 cases, participants then indicated which of the provided options most strongly influenced their prescribing decisions. Lastly, they completed 4 questions related to demographic factors that might influence their prescribing, e.g., about their "clinical grade". | 5 case vignette—uncomplicated UTI, severe pyelonephritis (other three were non UTI related) |
| [49] | Overtreatment of asymptomatic bacteriuria: a qualitative study | 21 Physicians, Switzerland | purposive sampling (21/ 69 = 30%) | cross-sectional qualitative case-based, semi-structured, individual interviews with open-ended questions | Asymptomatic bacteriuria | semi-structured interviews using vignettes with open-ended questions | case-based, semi-structured, individual interviews with open-ended questions | 6 interview questions which focused on pre-defined topic guide i.e determinants of starting antibiotic treatment for suspected UTI, knowledge of scientific evidence and resulting evidence-based guidelines and adherence thereto, and understanding the awareness of the concept of "asymptomatic bacteriuria" | 6 case-based, semi-structured, individual interviews with open ended questions to understand motivators for unnecessary antibiotic prescribing for ASB |
| [50] | Nurses' Knowledge, Perception, and Self-Confidence Regarding Evidence-Based Antibiotic Use in the Long-Term Care Setting | 63 nurses pre-education and 57 nurses post educations, USA | convenience sampling (63/ 140 = 45% pre-education and 57/140 = 41% post-education) | Cross-sectional quantitative pre and post email survey | Acute uncomplicated Urinary Tract Infections | online surveys with Likert scale response options were made available through workplace emails or terminals | five-point Likert scale: (5) always, (4) often, (3) sometimes, (2) rarely, (1) never | five antibiotic stewardship knowledge/ perception questions with three to nine sub-questions each, and five self-confidence questions. The post education survey contained these questions plus six questions asking level of satisfaction with the educational module, applicability to practice, and preferences for future educational modules | 5 items on knowledge/ perceptions (three to nine sub questions each), 1 self-confidence, 1 level of satisfaction with antibiotic stewardship |

*(Continued)*

**Table 5.** (Continued)

| Author(s), Year | Study title | Number of respondents and country | Sampling strategy (% response rate, where reported) | Study Design and method | Therapeutic area | Method of data collection | Question format | Number of questions | Number of items |
|---|---|---|---|---|---|---|---|---|---|
| [51] | Impact of the medical specialty on knowledge regarding multidrug-resistant organisms and strategies toward antimicrobial stewardship | 456 Medical specialists (urologists, internists, surgeons, gynaecologist), Germany | not reported (456/1061 = 43%) | Cross-sectional self-administered quantitative survey | MDRO infections including UTI | surveys with Likert scale response options distributed to staff by participating hospital administrators | 4-point Likert scale response: (1) very unconfident, (2) unconfident (3) confident (4) very confident and one item response | 35 items with an average of 4 to 7 items under each construct of knowledge, attitude, and perceptions | 35 items: 4-point scale on constructs assessment and 1-item answer on factors influencing prescribing |
| [52] | Rapid diagnostic testing in the management of urinary tract infection: Potentials and limitations | 91 Physicians, USA | not reported | cross-sectional quantitative case vignettes: in person and online administered | Uncomplicated cystitis, recurrent UTI, Immunosuppressed, no UTI, Pyelonephritis, Urethritis, no UTI | surveys were distributed and completed from August 2017 to March 2018 both in-person and online by U.S. physicians who treat UTI | Simulated case-vignettes of patients with complicated and uncomplicated UTIs consisting of three segments on patient history and physical exam, rapid testing | 5 case-vignettes of patients presenting with uncomplicated cystitis, recurrent UTI, immunosuppressed (no UTI), Pyelonephritis and urethritis with three segments on patient history and physical exam and diagnostic test results using identification and quantification and antimicrobial susceptibility testing (AST). | 5 Simulated case-vignettes of patients with complicated and uncomplicated UTIs consisting of three segments on patient history and physical exam, rapid testing |
| [53] | Overestimation Error and Unnecessary Antibiotic Prescriptions for Acute Cystitis in Adult Women | 100 Physicians and nurses, Canada | not reported (historical data so response rate not given but 1/231 cases was removed) | cross-sectional self-administered quantitative study | Acute cystitis | physicians observation of real patient scenarios and completion of surveys with visual analogue scales | real patient scenarios of women with symptoms suggesting uncomplicated UTIs; visual analogue scale to determine likelihood of a positive urine culture | not given—There was only 1 visual analogue scale question. It is unclear whether physicians were asked to consult the decision aid before making their decision. | Real patient scenarios of women with symptoms suggesting uncomplicated UTIs; visual analogue scale to determine likelihood of a positive urine culture |
| [54] | Urinary Tract Infections in the Elderly: A Survey of Physicians and Nurses | 373 Physicians and nurses, USA | convenience sampling (373/1900 = 20%) | cross-sectional quantitative survey, mailed | UTI and asymptomatic bacteriuria | mailed survey with multiple selection of responses | based on the selection from a list of the symptoms and patient conditions that determine when to begin an antibiotic and the conditions that require monitoring for asymptomatic bacteriuria, respectively | 10 symptoms, 5 patient conditions, 5 condition options that require monitoring of asymptomatic bacteriuria | 2 multiple choice questions on symptoms and patient conditions that necessitate antibiotic prescription and 1 multiple choice question on conditions that require monitoring for asymptomatic bacteriuria |
| [55] | Decision making by general practitioners in diagnosis and management of lower urinary tract symptoms in women | 6 General Practitioners, U.K | not reported | Cross sectional self-administered quantitative survey informed by real patient case scenario | Lower UTIs | physicians observation of real patient scenarios and completion of surveys which included open and closed-ended questions. | open ended questions and visual analogue scales from "not at all" to "very well" and attitude to consultation from "dismayed" to "pleased" | not given—There was only 1 visual analogue scale question. It is unclear whether physicians were asked to consult the decision aid before making their decision. | 54 real patient scenarios of women with lower UTIs followed by open ended questions and 2-point visual analogue scale |

(Continued)

Table 5. (Continued)

| Author (s), Year | Study title | Number of respondents and country | Sampling strategy (% response rate, where reported) | Study Design and method | Therapeutic area | Method of data collection | Question format | Number of questions | Number of items |
|---|---|---|---|---|---|---|---|---|---|
| [56] | A survey of the views and capabilities of community pharmacists in Western Australia regarding the rescheduling of selected oral antibiotics in a framework of pharmacist prescribing | 90 Pharmacists, Australia | not reported (90/240 = 38%) | cross-sectional quantitative survey and case vignette, mailed | 1 case scenario on UTI in pregnancy and 1 on acute pyelonephritis | mailed survey and vignettes | five-point Likert scale: e.g. (1) strongly agree (2) agree (2) neutral (3) disagree and (4) strongly disagree, demographics, statements of views on expanding the pharmacist's role in prescribing antibiotics. The case vignettes consisted of seven scenarios and the respondents were asked for their preferred treatment option, under the hypothetical assumption that they were permitted to prescribe oral antibiotics | 9 statements of views questions. 7 case vignettes with up to 4 questions for each case. | 5-point scale, questionnaire, and graded case vignettes |
| [57] | Why are antibiotics prescribed for asymptomatic bacteriuria in institutionalized elderly people? A qualitative study of physicians' and nurses' perceptions | 38 physicians and nurses, Canada | convenience sampling (38/44 = 86.4%) | cross-sectional qualitative focus group discussions | Asymptomatic bacteriuria | Open ended focus group discussions facilitated by a medical anthropologist | focus group discussions with open ended questions to generate discussion in 3 main areas: the decision to order a urine culture, the decision to order antibiotics and possible strategies to reduce the prescription of antibiotics for asymptomatic bacteriuria | not mentioned but a summary of issues and themes identified is given comprising of 6 issues around nurse of physician interpretation of bacteriuria as symptomatic in the presence of non-symptomatic cases, ordering of urine cultures for non-specific changes in residents status, the central role of the nurse in communicating nonspecific changes in the health status of a resident physician and family members, the difficult in eliciting info about symptoms in frail adults, uncertainty in managing positive urine cultures and concern of liability of nurses and physicians | 3 items (The decision to order a urine culture, the decision to order antibiotics and possible strategies to reduce the prescription of antibiotics for asymptomatic bacteriuria) |

**Table 6. Psychometric properties.**

| Author (s), Year | Construct assessed | Source(s) used to inform data collected i.e., instrument development | Validity | Reliability | Sampling strategy (% response rate, where reported) | Acceptability | feasibility (administrative costs to complete) | Educational impact (applicability of results in a practical context) |
|---|---|---|---|---|---|---|---|---|
| [44] | attitude and behaviour constructs are discovered and not assessed | not mentioned | not reported | not reported | not reported | response rate not reported, no pilot testing and incomplete items not reported | verbal investigation using simulated patient carried out without the observation of the pharmacist | study did not evaluate appropriateness of decision to prescribe |
| [45] | attitudes influencing prescribing decision which are discovered and not assessed | not reported | not reported | not reported | not reported | response rate not reported, no pilot testing and incomplete items not reported | Doctors on the wards were handed the questionnaire and supervised while they completed (unaided) | study's evaluation on appropriateness of decision to prescribe achieved using local hospital guidelines, the British National Formulary and British thoracic society |
| [46] | attitude towards dispensing antibiotics without prescription | previous literature and work of Hadi et al. [58] | face and content validated and pilot-tested | not reported | Convenience sampling (100%) | 100% response rate, pilot tested for understanding, incomplete items not reported | questionnaires distributed by data collectors for self-completion by pharmacists and after two months simulated patients who were trained to approach pharmacy staff visited presenting clinical scenario of UTI | study's evaluation on appropriateness of decision to prescribe achieved using law enforcement guidelines at country level |
| [47] | perceptions of antibiotic resistance and antibiotic prescribing in UTIs of which perceptions constructs are discovered and not assessed | not reported | validation not reported but interviews were pilot tested | not reported | Purposive sampling (not clear) | response rate not clear, no pilot testing and incomplete items not reported | researcher facilitated the interview at workplace | study's evaluation on appropriateness of decision to prescribe achieved using Swedish national guidelines |
| [48] | knowledge based on scientific evidence and influence on prescribing | Authors (based on a range of bacterial infections managed in the emergence department) | not reported | not reported | not reported (initial approached not given) | 65% response rate, no pilot testing and incomplete items not reported | self-administered survey completed online | study's evaluation on appropriateness of decision to prescribe achieved using the national antimicrobial prescribing survey guidelines |
| [49] | knowledge of scientific evidence, asymptomatic bacteriuria management concepts, and attitude and behaviour towards treatment of ASB | Authors (infectious disease fellow) and reviewed by infectious disease senior physicians, epidemiologist, and behavioural psychologist | face and content validated by infectious disease senior physicians, epidemiologist, and behavioural psychologist | not reported | purposive sampling, (30%) | 30% response rate, no pilot testing and incomplete items not reported | researcher facilitated the interview at workplace | study's evaluation on appropriateness of decision to prescribe achieved using national guidelines |

*(Continued)*

**Table 6.** (Continued)

| Author(s), Year | Construct assessed | Source(s) used to inform data collected i.e., instrument development | Validity | Reliability | Sampling strategy (% response rate, where reported) | Acceptability | feasibility (administrative costs to complete) | Educational impact (applicability of results in a practical context) |
|---|---|---|---|---|---|---|---|---|
| [50] | knowledge, attitude, perceptions, and self-confidence | Modified from the Minnesota Department of Health's Antibiotics Stewardship Program Toolkit for Long-Term Care Facilities | face and content validated using 21 post baccalaureate Doctors of Nurse practitioner students, 2 pharmacy students, 2 pharmacists, and 2 nurses working in long-term care setting | not reported | Convenience sample of 140 nurses, (response rate: 45% pre-education and 41% post-education) | 45% pre-education and 41% post-education response rate, pilot tested for understanding, incomplete items not reported | self-administered survey completed online | study's evaluation on appropriateness of decision to prescribe achieved using the Loeb minimum criteria for the initiation of antibiotics in long-term care residence |
| [51] | Knowledge, Attitudes, and perceptions | Authors who were experts in urology and medical sciences | face and content validated by infectious disease experts and pilot-tested by 15 clinicians representing urologists and non-urologists | not reported | not reported | 43% response rate reported, no pilot testing and incomplete items not reported | survey sent using a scanner to physicians who self-completed | study's evaluation on appropriateness of decision to prescribe achieved using national guidelines |
| [52] | knowledge of guidelines and treatment, attitudes for comfort with rapid testing | not reported | not reported | not reported | not given (81% of centres) | not reported | surveys posted on social media and contacting departmental chairs: completed online and in-person by physicians | study's evaluation on appropriateness of decision to prescribe achieved using the infectious Disease Society of America (IDSA) guidelines |
| [53] | knowledge of appropriateness of prescribing based on urine culture without the requirement of physicians to use decision aids in making their choice. | not reported | not reported | not reported | Convenience sampling (20%) | not reported (historical data so response rate not given but 1/231 cases was removed) | survey completed in person by physician at place of work following a real patient consultation | study's evaluation on appropriateness of decision to prescribe achieved using a retrospective audit using the actual test outcomes to assess the women's condition |
| [54] | physicians attitudes towards patient symptoms and conditions | not reported | not reported | not reported | | 20% response rate, no pilot testing and incomplete items not reported | survey mailed to physicians who self-completed | study's evaluation on appropriateness of decision to prescribe achieved using guidelines determined by the Internal Review Board of the University of North Dakota. |

(*Continued*)

**Table 6.** (Continued)

| Author (s), Year | Construct assessed | Source(s) used to inform data collected i.e., instrument development | Validity | Reliability | Sampling strategy (% response rate, where reported) | Acceptability | feasibility (administrative costs to complete) | Educational impact (applicability of results in a practical context) |
|---|---|---|---|---|---|---|---|---|
| [55] | knowledge of patient and likelihood of positive culture, attitude towards patient characteristics influencing prescribing decisions | developed by authors who were senior lectures in the department of health sciences and psychiatry, other parts of the questionnaire were modified based on the general health questionnaire to detect probable psychological disorder and the menstrual distress questionnaire. | not reported | not reported | not reported | response rate not reported, no pilot testing and incomplete items not reported | survey completed in person by physician at place of work following a real patient consultation | study's evaluation on appropriateness of decision to prescribe achieved using mid sample urine tests after end of patient assessment |
| [56] | attitudes and pharmacist views on down scheduling of selected antibiotics | developed by authors who are pharmacists and medical Doctors based on literature reviews and pilot testing with community pharmacists who have extensive antibiotic experience | face and content validated; pilot tested by 6 community pharmacists | not reported | | 38% response rate, pilot tested for understanding, incomplete items not reported | survey mailed to pharmacists who self-completed | study's evaluation on appropriateness of decision to prescribe achieved using the Australian Therapeutic Guidelines (ATG) for antibiotics and existing literature. |
| [57] | Perceptions, attitudes, and opinions in the ordering of urine cultures and the prescribing of antibiotics for asymptomatic bacteriuria in institutionalized elderly people. the perception, attitudes and opinions are constructs that are discovered and not assessed | not mentioned | not reported | not reported | Convenience sampling (76% response rate). | 86.4% response rate, no pilot testing and incomplete items not reported | focus group discussions: facilitated by a medical anthropologist | study did not evaluate appropriateness of decision to prescribe |

process. However, there was limited evidence on the actual validation process presented in the paper, in terms of items that were excluded from the final instrument.

**Reliability.** None of the studies provided information about the reliability of the instrument.

**Acceptability.** The acceptability of an instrument assesses the clinicians understanding of the instrument (Acceptability 1 in Table 8), level of completion of the instrument (Acceptability 2) and evidence of successful instrument completion within an appropriate setting (Acceptability 3) [38]. Few studies [46, 47, 51, 56] reported having pilot tested their instrument. We

**Table 7. Psychometric properties evaluated within each study using COSMIN and Terwee scores.**

| Author (s), Year | Face and Content validity | | | Construct validity | | | Criterion validity | | | Internal consistency | | | Reliability | | |
|---|---|---|---|---|---|---|---|---|---|---|---|---|---|---|---|
| | Result | COSMIN | Terwee | Result | COSMIN | Terwee | Result | COSMIN | Terwee | Result Cronbach's alpha | COSMIN | Terwee | Result | COSMIN | Terwee |
| [44] | - | - | - | - | - | - | - | - | - | - | - | - | - | - | - |
| [45] | | - | - | - | - | - | - | - | - | - | - | - | - | - | - |
| [46] | Face and content validated and pilot-tested (sample type not given) | doubtful | ? | - | - | - | - | - | - | - | - | - | - | - | - |
| [47] | validation not reported but interviews were pilot tested (sample type not given) | doubtful | ? | - | - | - | - | - | - | - | - | - | - | - | - |
| [48] | - | - | - | - | - | - | - | - | - | - | - | - | - | - | - |
| [49] | Face and content validated by infectious disease senior physicians, epidemiologist and behavioural psychologist | adequate | (+?) | - | - | - | - | - | - | - | - | - | - | - | - |
| [50] | Face and content validated using 21 post baccalaureate Doctors of Nurse practitioner students, 2 pharmacy students, 2 pharmacists, and 2 nurses working in long-term care setting | very good | (+?) | - | - | - | - | - | - | - | - | - | - | - | - |
| [51] | Face and content validated by infectious disease experts and pilot-tested by 15 clinicians representing urologists and non-urologists | very good | (+?) | - | - | - | - | - | - | - | - | - | - | - | - |
| [52] | - | - | - | - | - | - | - | - | - | - | - | - | - | - | - |
| [53] | - | - | - | - | - | - | - | - | - | - | - | - | - | - | - |
| [54] | - | - | - | - | - | - | - | - | - | - | - | - | - | - | - |
| [55] | - | - | - | - | - | - | - | - | - | - | - | - | - | - | - |
| [56] | Face and content validated; pilot tested by 6 community pharmacists | adequate | (+?) | - | - | - | - | - | - | - | - | - | - | - | - |
| [57] | - | - | | - | - | - | - | - | - | - | - | - | - | - | - |

+ positive, – negative,? Indeterminate (+?) mixed

**Table 8. Additional psychometric properties evaluated within each study adapted from Beattie et al. [41].**

| Author(s), Year | Acceptability | | | Feasibility | | | Educational impact | | |
|---|---|---|---|---|---|---|---|---|---|
| | 1 | 2 | 3 | 1 | 2 | 3 | 1 | 2 | 3 |
| [44] | - | - | - | - | - | **** | - | - | **** |
| [45] | - | - | - | - | - | **** | - | - | **** |
| [46] | **** | **** | **** | - | - | **** | - | - | **** |
| [47] | - | - | - | - | - | **** | - | - | **** |
| [48] | - | **** | **** | - | - | **** | **** | **** | **** |
| [49] | - | - | ** | - | - | **** | - | - | **** |
| [50] | **** | **** | **** | - | - | **** | - | - | **** |
| [51] | - | **** | - | - | - | **** | - | - | **** |
| [52] | - | - | - | - | - | **** | - | - | **** |
| [53] | - | - | ** | - | - | **** | - | - | **** |
| [54] | - | - | ** | - | - | **** | - | - | **** |
| [55] | - | - | - | - | - | **** | - | - | **** |
| [56] | **** | - | ** | - | - | **** | - | - | **** |
| [57] | - | **** | **** | - | - | **** | - | - | **** |

**Ratings of study quality:**

*poor

** fair

***good

****excellent

Beattie, M., Murphy, D. J., Atherton, I. & Lauder, W. 2015. Instruments to measure patient experience of healthcare quality in hospitals: a systematic review. *Systematic Reviews*, 4, 97.

classified this as evidence of participants' understanding of the instrument, thus having a very good score on this psychometric property. Although, Hale et al. [50] did not mention pilot testing of the instrument, a large sample of healthcare professionals (n = 27) were asked to assess the face validity assessment of the instrument, which was revised based on their feedback. Therefore, we classified this as evidence of pilot testing and assigned a very good utility score of subjects' understanding of the instrument items. Information related to non-response for individual items was generally not reported. However, three studies reported response rates less than 40% [49, 54, 56] and studies with a response rate less than 40% were scored as doubtful as per matrix criteria according to Beattie et al. [41]. Five studies did not indicate the response rates [44, 45, 47, 52, 55]. The study by McIsaac and Hunchak et al. [53] was based on historical data, so no response rate was given but patient case (out of 231) was removed. The five studies with a "very good" score [46, 48, 50, 51, 57] as they had response rates above 40%.

**Feasibility.** The feasibility of an instrument can be determined by how long it takes for participants to complete the instrument (Feasibility 1 in Table 8), the minimum number of observations required to reach the required level of reliability (Feasibility 2), and the administrative costs of completing the survey (Feasibility 3) [38]. None of the studies reported on the time taken to complete their instrument or minimum survey completion time therefore, the subcategories of feasibility were not evaluated. However, all the instruments were easily embedded withing the existing resource, e.g., clinicians' workplace, with minimal additional support required. Therefore, on this category all instruments had a very good score.

**Educational impact.** Educational impact was evaluated by consideration of the instruments' purpose (Education Impact 1 in Table 8), ease of translation of the scoring system (Education Impact 2) and the feedback of the results (Education Impact 3) [38]. An instrument

could be rated as having the potential of achieving high educational impact if it evaluated the respondents' decisions/scores using evidence-based guidelines, literature, or clinical findings. Amongst the included studies appropriateness of treatment decision was determined prospectively using various information such as local hospital guidelines, the British National Formulary, and the British Thoracis Society [45]; the country of study's national evidence-based guidelines [46–49, 51]; existing literature and country of study guidelines [56]; Loeb minimum criteria for the initiation of antibiotics in long term care residence [50]; the Infectious Disease Society of America's recommendations [52]; retrospective audit using actual test outcomes to assess the women's condition [53]; and Internal Review Board of the University of North Dakota [54]. Only one study's instrument had an easy to translate scoring system i.e., acceptable level of knowledge and/or attitudes [48].

### Instrument knowledge and attitude items assessed

The study purpose in relation to knowledge and attitudes assessed varied in the studies included as shown in Table 6 above. In 5 of the 14 studies, the study purpose did not relate to measuring knowledge and attitudes that influence prescribing of antibiotics for UTIs but instead to identifying these factors. The study by Alrasheedy et al. [46] measured attitudes towards dispensing antibiotics without a prescription for the treatment of UTIs.

Interestingly, when the instrument items were classified according to the construct of knowledge or attitude, the content of the instruments varied widely as shown in Table 9. A detailed content comparison of the constructs represented in the fourteen studies is provided in S6 Table. Overall knowledge was more commonly assessed than attitudes. The most common construct of knowledge assessed was "condition" (13/14 studies) and the least assessed was "scientific rationale" (8/14 studies). Additional influences on healthcare professionals' knowledge were identified in the instruments. These included knowledge of patient characteristics such as age, previous medical history (e.g., previous history of resistant pathogens, antibiotic intake, comorbidities), marital status, childbearing potential, and socioeconomic status, which were assessed and reported in 9 of the 14 studies. The most common construct of attitudes was "confidence" (5/14 studies) and the least assessed was "indifference" (0/14). The influence of healthcare professionals' attitude towards patient characteristics was also identified in the instruments. Patient characteristics that influence attitude included healthcare professionals' familiarity with patients and their attitude towards patient confidence in the treatment provided.

## Discussion

This systematic review is the first to assess the psychometric properties of available instruments used for the measurement of healthcare professionals' knowledge about and attitudes toward antibiotic prescribing for UTIs. It is also the first to synthesise the knowledge and attitude items assessed using a mixed-model theoretical framework. Fourteen instruments were identified for the evaluation and reporting of psychometric properties. Given that knowledge and attitudes are key determinants of prescribing behaviour among healthcare professionals, it is important that the instruments for assessing knowledge and attitudes are tested in varied settings and known to be robust enough for use with healthcare professionals who have a prescribing role in improving patent outcomes. Almost none of these instruments covered all important subcategories of knowledge, attitude, and/or patient characteristics that have been documented in the literature as influencing prescribing.

None of the identified studies reported on all of their instrument's psychometric properties. The instrument with the "best" measurement properties was that developed by Hale et al. [50].

**Table 9. Overall knowledge and attitude constructs assessed.**

| Author(s), Year | Knowledge | | | | | Attitudes | | | | | | |
|---|---|---|---|---|---|---|---|---|---|---|---|---|
| | Scientific rationale | Knowledge of condition | Procedural knowledge | Task of environment | Patient characteristics | Complacency | Fear | Ignorance | Indifference | Responsibility of others | Confidence | Patient characteristics |
| [44] | ✓ | ✓ | ✓ | ✓ | ✓ | ✓ | - | - | - | - | - | - |
| [45] | ✓ | ✓ | ✓ | ✓ | ✓ | - | - | - | - | ✓ | - | - |
| [46] | ✓ | ✓ | - | ✓ | ✓ | - | ✓ | ✓ | - | - | ✓ | - |
| [47] | ✓ | ✓ | ✓ | ✓ | - | ✓ | ✓ | - | - | ✓ | ✓ | ✓ |
| [48] | - | ✓ | ✓ | ✓ | ✓ | - | - | - | - | - | - | - |
| [49] | ✓ | ✓ | ✓ | - | - | ✓ | - | - | - | - | - | - |
| [50] | ✓ | ✓ | ✓ | ✓ | ✓ | - | - | - | - | - | - | - |
| [51] | ✓ | - | - | ✓ | ✓ | - | - | ✓ | - | ✓ | ✓ | - |
| [52] | ✓ | ✓ | ✓ | ✓ | ✓ | - | - | - | - | - | ✓ | - |
| [53] | - | ✓ | ✓ | - | ✓ | - | - | - | - | - | - | - |
| [54] | - | ✓ | - | - | - | - | - | - | - | - | - | - |
| [55] | - | ✓ | ✓ | ✓ | ✓ | ✓ | - | - | - | - | - | - |
| [56] | - | ✓ | - | - | - | - | - | - | - | ✓ | ✓ | ✓ |
| [57] | - | ✓ | ✓ | - | - | - | - | - | - | - | - | - |
| **Total Number of Studies (N)** | 8 | 13 | 10 | 9 | 9 | 4 | 2 | 2 | 0 | 3 | 5 | 2 |

There was strong evidence that their instrument satisfied the requirements for face and content validity. Face and content validity were assessed and reported in just 5 of the 14 studies. None of the studies reported a detailed process or statistical analysis of validation on their instruments. Only two studies [51, 56] conducted pilot studies as part of the validation process [59]. None of the studies reported on the reliability of the instruments, which is a prerequisite to assuring the integrity and quality of a measurement instrument [60]. Therefore, it is uncertain whether any of the instruments capture the true state of healthcare professionals' knowledge and attitudes around prescribing antibiotics for UTIs.

The acceptability of the instruments was low as only five studies reported response rates above 40%. The feasibility of most instruments was achieved because minimal administrative costs were incurred when completing the instruments. This is because most instruments were self-administered within the clinicians' workplace setting. However, none of the studies reported the time taken to complete the instruments thereby questioning this aspect of feasibility (time cost to participant). In terms of the educational impact, only one study's instrument had a clear purpose and an easy to translate scoring system (i.e., what constitutes good knowledge and/or attitude) [48]. The feedback from the results was validated using the appropriateness of the decision to prescribe for all studies. However, it was not reported whether this feedback was communicated to the respondents, which means the full impact of the instruments towards feedback provision in addressing inappropriate behaviours/misconceptions is questionable.

Concerning what aspects of knowledge and attitude the instruments measured, three [44, 50, 52] intended to assess all of the knowledge constructs, and none intended to assess all of the attitude constructs. Other instruments also identified one or more concepts related to the knowledge and attitude constructs without explicitly stating an intention to measure them [44–47, 57]. Given the limited psychometric evidence for the included instruments, we cannot provide a definitive, evidence-based recommendation for a particular instrument to measure healthcare professionals' knowledge about and attitudes toward antibiotic prescribing for UTIs.

The findings reported in this review show that there are no instruments for which there is evidence of all of the required psychometric properties of validity, reliability, acceptability, feasibility, and potential for educational impact to enable adequate measurement of healthcare professionals' knowledge and attitudes towards the use of antibiotics for the treatment of UTIs. Measurement instruments with satisfactory psychometric properties are required to yield useable findings. Without such useable measures of knowledge and attitudes, it is impossible to determine if such knowledge and attitudes require improvement and if so to plan and evaluate appropriate ameliorative interventions.

The findings reported in this review also support the need for solid theoretical foundations to define the constructs of knowledge and attitude from which such instruments need to be developed. Previous researchers of similar studies have reported a variation of constructs used for measurement instruments addressing inappropriate antibiotic use [61]. Varying definitions of knowledge and attitudes exist [17, 29] and some researchers fail to define the constructs measured [62]. Therefore, it is essential to ensure all constructs are underpinned by a sound theoretical framework to ensure all relevant items of knowledge and attitude constructs are considered for inclusion in future instruments. The use of cross-theory constructs has been reported in healthy psychology studies as a more appropriate/effective approach for understanding variation in prescribing behaviour than using single theory constructs [63]. The proposed model used in this systematic review offers a potential model as a basis for use in research. The choice of this model was not to employ a framework based on empirical affirmation but instead one that offered a pragmatic approach in the design of future behavioural change interventions [64].

## Strengths and limitations

The current systematic review excluded studies not published in English and this is a limitation. This systematic review did not include unpublished studies, which may have contained evidence of negative results about the measurement properties of the instruments. However, the inclusion of only peer-reviewed studies may have enhanced the quality of the studies included. The research is also unique in that the instrument items were synthesised by determining if they assessed knowledge and attitudes according to theoretically underpinned constructs of knowledge and attitudes, thereby, attempting to achieve construct validity. However, due to the heterogeneity of the stated aims of different studies, construct validity may have not been fully achieved. The methods employed in this systematic review present as the first step for future work on studies that aim to measure knowledge and attitudes. In line with the aims of this review, the "practice" component of the WHO's knowledge, attitudes, and practice framework was excluded. Future studies could explore healthcare professionals' practices–i.e., their actual prescribing of antibiotics for UTIs, and link practices with appropriate measures of knowledge and attitude.

## Conclusion

The present review highlights the lack of evidence for the psychometric properties of instruments used in assessing knowledge and attitudes among clinicians towards the treatment and management of UTIs. Numerous gaps were identified in the evidence for the identified instruments. The instrument with the best properties [50] assessed face and content validity with a justifiable number of subject matter experts whose roles were clearly stated, and details of pilot testing were provided. However, even this study did not sufficiently report on other psychometric properties such as reliability and educational impact. Therefore, we were unable to make strong recommendations concerning which instruments ought to be used to measure clinicians' knowledge and attitude towards the use of antibiotics for the treatment and management of UTIs. This review may provide a starting point for researchers to identify instruments that are currently used for this purpose among clinicians and should lead to further work to provide evidence for all measurement properties.

## Supporting information

**S1 Fig. Pathogenesis of UTI.**
(TIF)

**S2 Fig. Physician prescribing behaviour model.**
(TIF)

**S3 Fig. PRISMA checklist.**
(PDF)

**S4 Fig. Search strategy.**
(TIF)

**S1 Table. Inclusion and exclusion criteria.**
(PDF)

**S2 Table. Definition of study terms.**
(ODT)

**S3 Table. Additional questions using adopted Beattie et al. [41] matrix.**
(XLS)

**S4 Table. Quality of psychometric results Terwee et al. [37].**
(PDF)

**S5 Table. Kappa score.**
(PDF)

**S6 Table. Detailed comparison of knowledge and attitudes assessed.**
(PDF)

## Acknowledgments

We thank Samantha Johnson (Information Specialist) for developing and supporting the screening of the searches.

## Author Contributions

**Conceptualization:** Angela Kabulo Mwape, Kelly Ann Schmidtke, Celia Brown.

**Data curation:** Angela Kabulo Mwape.

**Formal analysis:** Angela Kabulo Mwape, Kelly Ann Schmidtke, Celia Brown.

**Funding acquisition:** Kelly Ann Schmidtke, Celia Brown.

**Investigation:** Angela Kabulo Mwape, Celia Brown.

**Methodology:** Angela Kabulo Mwape, Kelly Ann Schmidtke, Celia Brown.

**Project administration:** Angela Kabulo Mwape, Kelly Ann Schmidtke, Celia Brown.

**Resources:** Angela Kabulo Mwape, Celia Brown.

**Software:** Angela Kabulo Mwape.

**Supervision:** Kelly Ann Schmidtke, Celia Brown.

**Validation:** Kelly Ann Schmidtke, Celia Brown.

**Visualization:** Angela Kabulo Mwape.

**Writing – original draft:** Angela Kabulo Mwape.

**Writing – review & editing:** Angela Kabulo Mwape, Kelly Ann Schmidtke, Celia Brown.

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
