## [Decision Letter · Decision Letter 0]

19 Jan 2022

PONE-D-21-35085Instruments used to measure knowledge and attitudes of healthcare professionals towards antibiotic use for the treatment of urinary tract infections: A systematic review.PLOS ONE

Dear Dr. Mwape,

Thank you for submitting your manuscript to PLOS ONE. After careful consideration, we feel that it has merit but does not fully meet PLOS ONE’s publication criteria as it currently stands. Therefore, we invite you to submit a revised version of the manuscript that addresses the points raised during the review process.

We look forward to receiving your revised manuscript.

Kind regards,

Vijayaprakash Suppiah, PhD

Academic Editor

PLOS ONE

Journal Requirements:

2. Thank you for stating the following in the Financial Section of your manuscript:

“The current project was supported by the National Institute for Health Research (NIHR) Applied Research Centre (ARC) West Midlands, grant number NIHR200165. The views expressed are those of the author(s) and not necessarily those of the NIHR, ARC, or the Department of Health and Social Care. The funders had no role in the design of the study and collection, analysis, and interpretation of data and in writing the manuscript.”

Please note that funding information should not appear in other areas of your manuscript. We will only publish funding information present in the Funding Statement section of the online submission form.

“The current project was supported by the National Institute for Health Research (NIHR) Applied Research Centre (ARC) West Midlands, grant number NIHR200165. The views expressed are those of the author(s) and not necessarily those of the NIHR, ARC, or the Department of Health and Social Care. The funders had no role in the design of the study and collection, analysis, and interpretation of data and in writing the manuscript.”

Reviewers' comments:

Reviewer's Responses to Questions

**Comments to the Author**

1. Is the manuscript technically sound, and do the data support the conclusions?

Reviewer #1: Partly

2. Has the statistical analysis been performed appropriately and rigorously? 

Reviewer #1: N/A

3. Have the authors made all data underlying the findings in their manuscript fully available?

Reviewer #1: Yes

4. Is the manuscript presented in an intelligible fashion and written in standard English?

Reviewer #1: Yes

5. Review Comments to the Author

Reviewer #1: The systematic review entitled “Instruments used to measure knowledge and attitudes of healthcare professionals towards antibiotic use for the treatment of urinary tract infections: A systematic review” has highlighted the important aspects of the knowledge and attitudes of healthcare professionals towards antibiotic use for the treatment of urinary tract infections. The following are my comments for the Authors.

1. Overall, the present review if of great importance but the authors missed the main important aspect of KAP studies in relation to the healthcare professionals and that’s “Practices” of antibiotics for UTIs. As along with knowledge, attitude; practices are the prime factor in the KAP models studies and measurement. A reasonable explanation is needed.

2. The introduction must be started from the very basic definition of the UTIs and the cause of UTIs after that the epidemiology for UTI may include.

3. The scale of measurement for the Psychometric approach must be clear in the methods section and how it was employed in the inclusion criteria. How was the PICO model fit in the inclusion criteria and on which basis the studies were included, either on PICO or Psychometric measurement?

4. Were the included studies were finalized after the risk biased assessment? Is any scale was used for the risk bias assessment and the protocols at all.

5. The main evidence which has been drawn from the present review is not cleared. Very specific evidence must be mentioned that must represent the whole picture of the review. Such measuring scales and other factors must be controlled and the conclusion and the evidence from the review must be cleared that would be helpful for the readers and researchers in the field.

6. Recent references are needed to be included in the introduction and discussion parts.

Wish you all the best.

Thanks, and Best Regards.,

6. PLOS authors have the option to publish the peer review history of their article (what does this mean?). If published, this will include your full peer review and any attached files.

Reviewer #1: **Yes: **Faiz Ullah Khan

---

## [Author Response · Author response to Decision Letter 0]

3 Mar 2022

Warwick Medical School

University of Warwick

Medical School Building, 

Coventry. CV4 7HL

United Kingdom

Tel: + 44 (0)24 7657 4880

E-mail: Angela.Mwape@warwick.ac.uk

Dear Vijayaprakash Suppiah, PhD. Academic editor,

Thank you for inviting us to revise our manuscript titled " Instruments used to measure knowledge and attitudes of healthcare professionals towards antibiotic use for the treatment of urinary tract infections: A systematic review” (PONE-D-21-35085)

The reviewer comments were very helpful, and many changes were made. Often these changes are highlighted as yellow in the manuscript. However, some changes are too large to simply highlight. We cannot highlight changes that were the result of cutting and heavy restructuring. 

A revision letter is attached in which we describe how we responded to the editor’s and reviewers’ comments. Where possible we state where changes were made in the manuscript (with page and line numbers). All authors have approved these changes. 

Thank you very much for your re-consideration. 

Sincerely Yours,

Miss Angela Mwape on behalf of all co-authors

---

## [Decision Letter · Decision Letter 1]

6 Apr 2022

Instruments used to measure knowledge and attitudes of healthcare professionals towards antibiotic use for the treatment of urinary tract infections: A systematic review.

PONE-D-21-35085R1

Dear Dr. Mwape,

We’re pleased to inform you that your manuscript has been judged scientifically suitable for publication and will be formally accepted for publication once it meets all outstanding technical requirements.

Kind regards,

Vijayaprakash Suppiah, PhD

Academic Editor

PLOS ONE

Reviewers' comments:

Reviewer's Responses to Questions

**Comments to the Author**

1. If the authors have adequately addressed your comments raised in a previous round of review and you feel that this manuscript is now acceptable for publication, you may indicate that here to bypass the “Comments to the Author” section, enter your conflict of interest statement in the “Confidential to Editor” section, and submit your "Accept" recommendation.

Reviewer #1: All comments have been addressed

2. Is the manuscript technically sound, and do the data support the conclusions?

Reviewer #1: Yes

3. Has the statistical analysis been performed appropriately and rigorously? 

Reviewer #1: N/A

4. Have the authors made all data underlying the findings in their manuscript fully available?

Reviewer #1: Yes

5. Is the manuscript presented in an intelligible fashion and written in standard English?

Reviewer #1: Yes

6. Review Comments to the Author

Reviewer #1: All the comments have addressed by the authors.

I have no specific comments at this stage, but the authors may improve and look to the English editing along with the grammar.

Good luck!

7. PLOS authors have the option to publish the peer review history of their article (what does this mean?). If published, this will include your full peer review and any attached files.

Reviewer #1: **Yes: **Faiz Ullah Khan

---

## [Editor Report · Acceptance letter]

11 May 2022

PONE-D-21-35085R1 

Instruments used to measure knowledge and attitudes of healthcare professionals towards antibiotic use for the treatment of urinary tract infections: A systematic review.

Dear Dr. Mwape:

I'm pleased to inform you that your manuscript has been deemed suitable for publication in PLOS ONE. Congratulations! Your manuscript is now with our production department. 

Kind regards, 

on behalf of

Dr. Vijayaprakash Suppiah 

Academic Editor

PLOS ONE